# Genome-Wide Identification and Expression Profiling of B3 Transcription Factor Genes in *Prunus armeniaca*

Xiaodan Shi [1,†], Wanwen Yu [1,†], Lin Wang [2,3,4,†], Han Zhao [2,3,4], Jingjing Hu [5,6,7], Tana Wuyun [2,3,4] and Huimin Liu [2,3,4,*]

1   Co-Innovation Center for the Sustainable Forestry in Southern China, Nanjing Forestry University, Nanjing 210037, China; sxdamber@163.com (X.S.); youeryuww@163.com (W.Y.)
2   State Key Laboratory of Tree Genetics and Breeding, Research Institute of Non-Timber Forestry, Chinese Academy of Forestry, Zhengzhou 450003, China; wanglin1815@163.com (L.W.); zhaohan@caf.ac.cn (H.Z.); tanatanan@163.com (T.W.)
3   Kernel-Apricot Engineering and Technology Research Center of National Forestry and Grassland Administration, Zhengzhou 450003, China
4   Key Laboratory of Non-Timber Forest Germplasm Enhancement and Utilization of National Forestry and Grassland Administration, Zhengzhou 450003, China
5   Department of Epidemiology, School of Public Health, Fudan University, Shanghai 200032, China; perfect2009@126.com
6   Key Laboratory of Public Health Safety, Ministry of Education, Fudan University, Shanghai 200032, China
7   Shanghai Pinnacles Medical Technology Co., Ltd., Shanghai 200126, China
*   Correspondence: luckyliuhm@163.com; Tel.: +86-17638183691
†   These authors contributed equally to this work.

**Abstract:** The B3 superfamily, which belongs to the plant-specific transcription factors, is widely involved in multiple biological processes. In apricot (*Prunus armeniaca*), the classification, structure, and function of the B3 superfamily are not yet clear. In this study, a total of 75 B3 genes were identified from the apricot genome. The apricot B3 superfamily can be divided into five subfamilies, i.e., REM, ARF, ABI3, RAV, and HSI, and genes in the same subfamily have similar structures. The distribution of B3 genes on chromosomes presents a clustering phenomenon. Tandem duplication is the main mode of apricot B3 family expansion, and gene duplication mainly occurs in the REM and ARF subfamilies. Many B3 genes originated from a common ancestor of *Arabidopsis* and apricot before lineage divergence, and ancestor genes expanded fewer times in apricot than in *Arabidopsis*. Gene ontology analysis showed that apricot B3 genes were closely related to vernalization, flower development, and shoot system development. *PaABI3-1* and *PaABI3-2* might play a positive regulation role in the transcription of *PaOleosin*, which encodes a lipid body protein. This study lays a foundation for the further study of the B3 superfamily function in apricot, especially the specific functions of the ABI3 subfamily in apricot kernel oil storage.

**Keywords:** apricot; B3 superfamily; transcription factor; gene expression; kernel development; oil storage

## 1. Introduction

Apricot (*Prunus armeniaca* L., Rosaceae) is one of the most delicious and commercially traded fruits in the world [1]. In addition to being consumed as a fruit, the kernel of apricots is widely utilized for edible, cosmetic, and medicinal purposes. The oil extracted from apricot kernel is rich in unsaturated fatty acids, oleic acids, phenols, and tocopherol, and it has high nutritional value [2]. Transcription factors are defined as transcription regulators that act by binding to specific *cis*-regulatory elements present in target gene promoters [3]. Some transcription factors are specific to plants, such as AP2-ERF, WRKY, and B3 families, whereas other transcription factors are also found in animals and yeast [4]. The studies of transcription factors in *Prunus* genus have mainly been performed in peach (*P. persica*)

but less in other species, including apricot [5]. To our knowledge, the research on the B3 transcription factor superfamily in apricot has not been reported.

The B3 transcription factor superfamily consists of a group of members containing one or several B3 DNA-binding domains (DBDs) [6]. The B3 domain was initially characterized in the maize gene *VIVIPAROUS1* (*VP1*) [7]. It is composed of amino acid residues with an average length of 100, and its secondary structure includes seven β-sheets and two α-helixes [8,9]. Many transcription factors containing DBDs are involved in plant vegetative and reproductive development, stress resistance induction, and hormone-related signal transduction [10,11]. Research on the B3 superfamily has been carried out in some model plants and cash crops, such as *Arabidopsis thaliana* [6,9], *Oryza sativa* (rice) [12], and citrus [13].

The B3 transcription factor superfamily can be divided into five subfamilies, including the ABI3/VP1, HSI (High-level expression of sugar inducible gene), ARF (Auxin Response Factor), RAV (Related to ABI3/VP1), and REM (Reproductive Meristem) families [14–18]. Proteins from different subfamilies have different structures and DNA binding sites. All ABI3 proteins have four highly conserved domains, including three basic domains: B1, B2, and B3 [19]. Among them, the B3 domain is necessary for the ABI3 protein to combine with the RY motif (DNA motif CATGCA (TG)) in many seed specific promoters. The HSI subfamily was once considered a VAL subgroup under the LAV (LEC2-ABI3-VAL) subfamily, but experiments by some researchers subsequently showed that HSI was significantly different from other members of the LAV subfamily [15,20]. Therefore, HSI seemed to be a new subfamily. The HSI protein contains four domains—namely, the B3, CW-like zinc finger, EAR (ethylene response factor [ERF]-associated amphiphilic repression) motif, and PHD (plant homeodomain)-like domains [21]. Each member of the ARF subfamily contains an amino-terminal B3 DBD, which can specifically bind with TGTCTC auxin response elements (AuxREs) that are found in promoters of early auxin response genes [22,23]. Most ARF proteins contain a middle region (MR) that activates or inhibits transcription and a C-terminal dimerization domain (CTD) that is related to motifs III and IV found in Aux/IAA proteins [24,25]. There are 13 members of the RAV subfamily in *Arabidopsis*, and they contain a C-terminal B3 domain, which can recognize a consensus CACCTG sequence. Six of the members additionally contain an N-terminal AP2 domain that recognizes the CAACA sequence [17,26]. The members of the REM subfamily contain at least one B3 DBD, usually contain two to three, and can contain up to seven [27].

In addition to structure, there are also some differences in the functions of various subfamilies. ABI3 can regulate the development of mature seeds [28]. For example, ABI3 is able to promote the accumulation of chlorophyll and anthocyanin during seed embryogenesis, and it is also able to facilitate the storage of oil and starch [29–31]. The effect of ABI3 on seed is also reflected in promoting seed dormancy [32]. In addition, ABI3 is involved in plant resistance to abiotic stresses. It mediates the dehydration stress response in plants by regulating the expression of several downstream genes, modulates low-temperature-induced freezing tolerance, and activates the expression of high-temperature-inducible genes to inhibit seed germination [33–35]. HSI can repress seed maturation and lipid biosynthesis, silence seed dormancy, and enable subsequent germination and seedling growth [36,37]. Members of the ARF subfamily are widely involved in auxin mediated physiological processes, including embryo patterning, vascular differentiation, leaf senescence, flower morphogenesis and organ abscission, apical dominance, lateral root formation, shoot elongation, fruit development, and abiotic stress response [38–45]. They are also involved in signal pathways related to hormones, such as ethylene, brassinosteroid, and GA [46]. RAV proteins can inhibit hypocotyl elongation, cause plant dwarfing, regulate seed size, and participate in the resistance of plants to biotic and abiotic stresses [47–49]. At present, there are a few studies on the functions of the REM subfamily, mainly focusing on vernalization response, female and male gametophyte development, and floral organ identity [27,50–52].

In this study, the B3 superfamily in apricot was analyzed at a genome-wide level. The members of the B3 superfamily in apricot were identified, and their characteristics were

analyzed. The phylogenetic relationship of B3 transcription factors between apricot and *Arabidopsis* was studied. In addition, the structures, conserved motifs, and domains of B3 genes were analyzed, and their positions on the chromosomes were found. The duplication events of B3 genes in apricot were examined, and the synteny between apricot and *Arabidopsis* was analyzed. In order to explore the potential functions of the B3 superfamily in the growth and development of apricot, *cis*-acting elements prediction, and gene ontology (GO) analysis were carried out. To investigate the expression patterns of apricot B3 genes in different tissues and developmental stages, we calculated the expression levels of B3 genes using RNA-seq data, and we then validated them through RT-qPCR experiments. Furthermore, we analyzed the correlation between the expression level of *ABI3* and oil content as well as *ABI3* and *Oleosin*. This study lays a foundation for the further study of the B3 superfamily function in apricot.

## 2. Materials and Methods

### 2.1. Identification and Analysis of B3 Superfamily Members in Apricot

To identify proteins containing the B3 domain in apricot, the Hidden Markov Model (HMM) profile of the B3 domain (PF02362) was downloaded from InterPro database (https://www.ebi.ac.uk/interpro/, accessed on 2 August 2022), and it was used to search protein sequences containing the B3 domain in the *P. armeniaca* genome (Sungold, http://apricotgpd.com/, accessed on 2 August 2022) through HMMER software (default parameters). The protein sequences of *Arabidopsis thaliana* (ver. Araport11) were then downloaded from the TAIR database (https://www.arabidopsis.org/, accessed on 1 September 2022). Taking *Arabidopsis* B3 protein sequences as queries, we built a local database by using the genome data of *P. armeniaca*, and we conducted a BlastP search with default parameters to screen the orthologs. The screened results of HMMER and BlastP were submitted to NCBI CDD (https://www.ncbi.nlm.nih.gov/cdd, accessed on 8 September 2022) to verify the completeness of B3 domains. The remaining genes were considered members of the B3 superfamily genes in apricot. The length, putative isoelectric points (PIs), molecular weights, instability index, aliphatic index, and grand average of hydropathicity (GRAVY) of apricot B3 TFs were predicted by using ExPASy (http://expasy.org//protparam/, accessed on 19 September 2022).

### 2.2. Multiple Sequence Alignments and Phylogenetic Analysis

Multiple sequence alignments of the B3 protein sequences from *Arabidopsis* and apricot were performed by using MUSCLE (version 5.1) with default parameters [53]. The phylogenetic tree was constructed by using the Neighbor-Joining (NJ) method on TreeBeST (version 1.9.2) with 1000 bootstrap replicates and by using the Maximum Likelihood (ML) method on IQ-tree (version 2.0.3) with the best-fit model of JTT+R8, visualized in iTOL (https://itol.embl.de/, accessed on 10 July 2023) [54–56].

### 2.3. Gene Structure, Conserved Motif, and Domain Analysis

The structures of the B3 genes were analyzed using TBtools [57]. The conserved motifs of B3 TFs were analyzed by using TBtools [57]. The number of motifs searched were set up to 20, and their widths were between 6 to 200. The analysis results were visualized using TBtools [57]. The domains of B3 TFs were identified by using NCBI Batch CD-Search (https://www.ncbi.nlm.nih.gov/Structure/bwrpsb/bwrpsb.cgi, accessed on 20 September 2022), and they were visualized using TBtools [57].

### 2.4. Chromosomal Locations and Synteny Analysis

The physical locations of apricot B3 genes were obtained from the gff file of *P. armeniaca* genome (v1.0, http://apricotgpd.com/, accessed on 26 September 2022). The distribution of B3 genes on chromosomes was visualized by using MapChart software [58]. The Multiple Collinearity Scan toolkit (MCScanX) was used to check the duplication events of

B3 genes [59], and the Dual Synteny Plot in TBtools was used to plot the synteny analysis figure of apricot and *Arabidopsis* [57].

### 2.5. Cis-Acting Elements and Gene Ontology (GO) Analysis

The 2 kb upstream sequence of each B3 gene was extracted from the *P. armeniaca* genome in order to analyze the *cis*-acting elements by using PlantCARE (http://bioinformatics.psb.ugent.be/webtools/plantcare/html/, accessed on 20 September 2022) [60]. Elements were visualized using TBtools [57]. Gene functions were annotated using eggNOG-mapper (http://eggnog-mapper.embl.de/, accessed on 20 November 2022), and GO enrichment of ORA mode was analyzed using TBtools [57].

### 2.6. Expression Analysis Using RNA-Seq Data and Verification Using RT-qPCR

To investigate the expression patterns of B3 genes, we used the apricot RNA-seq data obtained from the Genome Sequence Archive (PRJCA001987) to calculate the FPKM (fragments per kilobase of exon per million fragments mapped) values of B3 and ABI3 genes in different tissues, including leaf, flower bud, flower, kernels (K1–K5), and flesh (F1–F8) of five and eight different development stages, respectively. The clean reads from transcriptomes of the abovementioned tissues were mapped to the apricot reference genome using Hisat2 (version 2.2.4) [61] with default parameters. The read counts of genes were quantified using htseq-count software, as suggested previously [62]. Then, FPKM values were calculated in the R package using the following formula: FPKM = total exon fragments/(mapped reads (millions) × exon length (kb)). Heatmaps were generated in the R package based on normalized FPKM values. Hierarchical cluster analysis was performed based on the 'complete' clustering method.

To verify the expression levels obtained from RNA-seq data, we also conducted RT-qPCR experiments. Apricots (Shenyangdashuaifu) were grown located at Yuanyang Long-Term Experimental Base of Research Institute of Non-Timber Forestry in Yuanyang County, Henan Province, China. Kernels at 50, 60, 70, 80, and 90 (maturity) days after flowering (DAF) were collected and stored at $-80\,^\circ$C. Total RNA extraction and RT-qPCR were performed on a CFX96 Touch Real-Time PCR Detection System (BIO-RAD, Hercules, CA, USA) company according to the method described by Liu et al. [63]. Primer sequences are listed in Table S1. *UBQ* was used as the internal control. RT-qPCR was performed with three biological replicates per gene. Expression levels were calculated using the $2^{-\Delta\Delta Ct}$ method [64]. The correlation analysis between expression levels of RNA-seq and RT-qPCR was conducted using GraphPad Prism 8 based on the $\log_2$(fold change) of FPKM values and RT-qPCR results.

### 2.7. Oil Content Detection

Apricot (Shenyangdashuaifu) kernels were collected at 50, 60, 70, 80, and 90 DAF, and were then freeze-dried. The oil content of lyophilized apricot kernels was detected as suggested by Deng et al., 2021 [65].

## 3. Results

### 3.1. Identification of B3 Superfamily Genes in Apricot

Through screening of HMM and BlastP search and further confirmation of NCBI CDD, 75 genes in apricot were identified as B3 superfamily genes (Table S2). The lengths of their corresponding proteins ranged from 150 to 1166 (aa), their molecular weights varied from 17.4 to 130.5 kDa, and their PIs ranged from 4.54 to 10.02. The instability index, aliphatic index, and GRAVY of proteins ranged from 33.1 to 82.2, 49.2 to 87.51, and $-1.155$ to $-0.188$, respectively.

### 3.2. Phylogenetic Analysis 3.2 Phylogenetic of Apricot B3 Genes

To reveal the phylogenetic relationship between apricot B3 genes, 75 apricot B3 protein sequences and 87 *Arabidopsis* B3 protein sequences were used for multiple sequence

alignments, and then phylogenetic trees were constructed by the ML and NJ method (Figures 1 and S1). The subfamily classification of these two methods were consistent, except *PaRAV6*, which was in the HSI branch, according to the NJ method. Compared to the NJ method, a phylogenetic tree constructed by the ML method had a strong bootstrap value. Thus, we used the ML tree for subfamily classification. According to the ML phylogenetic tree, the B3 superfamily genes of apricot were divided into five subfamilies, and these genes were named according to different subfamilies. REM was the largest subfamily, including 42 B3 genes, followed by ARF with 17 members. The RAV family contained eight B3 genes. The RAV and HSI families were small subfamilies, with only four B3 genes for each.

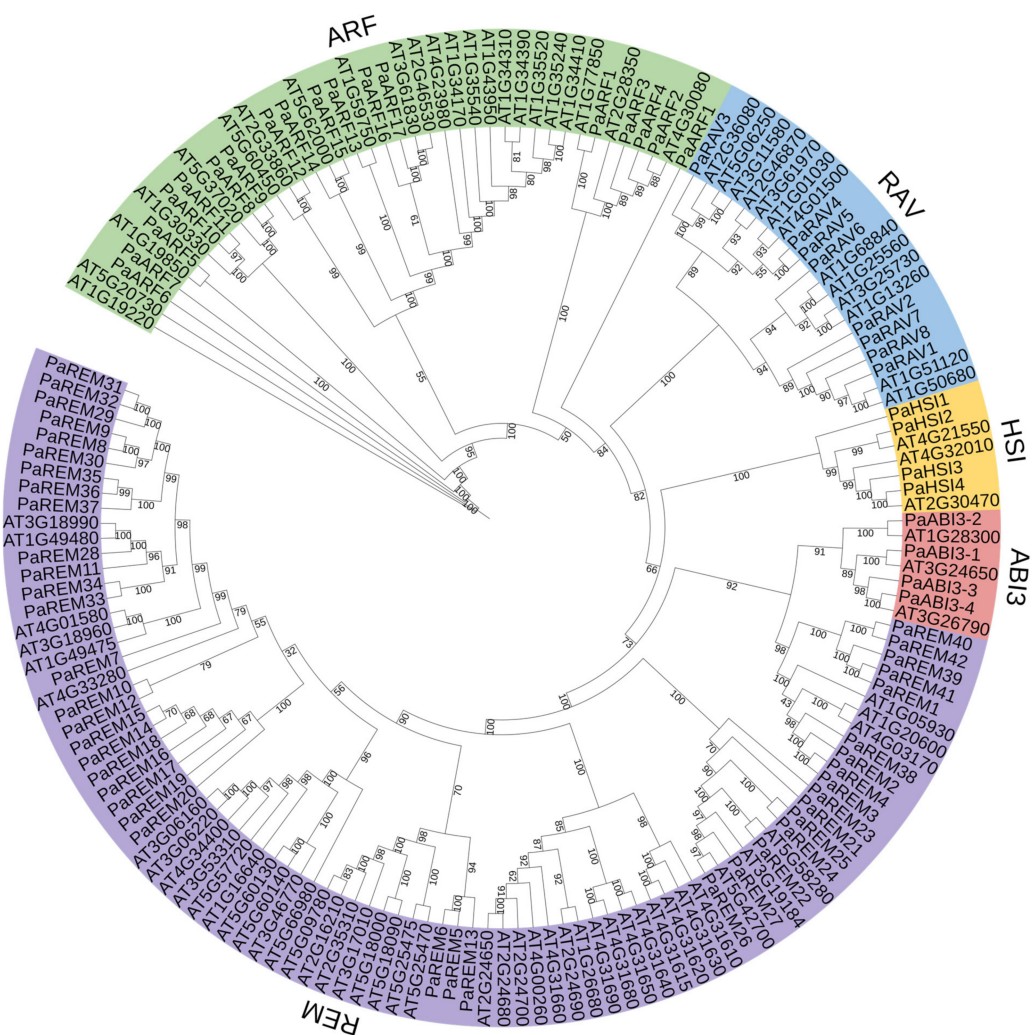

**Figure 1.** The phylogenetic tree of B3 family members. The tree was constructed based on the ML method using IQ-Tree and visualized in iTOL. Five color blocks represent five subfamilies. The number on the tree branches represents the bootstrap value.

### 3.3. Gene Structure, Conserved Motif, and Domain Analysis of Apricot B3 Genes

To explore the structural diversity and evolutionary relationship of apricot B3 genes, the exon–intron structure map was drawn (Figure 2). The apricot B3 genes had 1–15 exons. The members of ABI3 subfamily contained 5–7 exons. Four members of the HSI subfamily contained 10–14 exons. The number of exons in the eight RAV genes was fewer than that in other subfamilies, in the range of 1–4. In the ARF subfamily, there were 3–5 exons in *PaARF1* to *PaARF4* and 12–15 exons in *PaARF5* to *PaARF17*. Consistent with the phylogenetic tree, ARF genes with 3–5 and 12–15 exons were clustered into two large branches,

respectively. The REM subfamily members contained 1–10 exons, representing the most various structures among the five subfamilies.

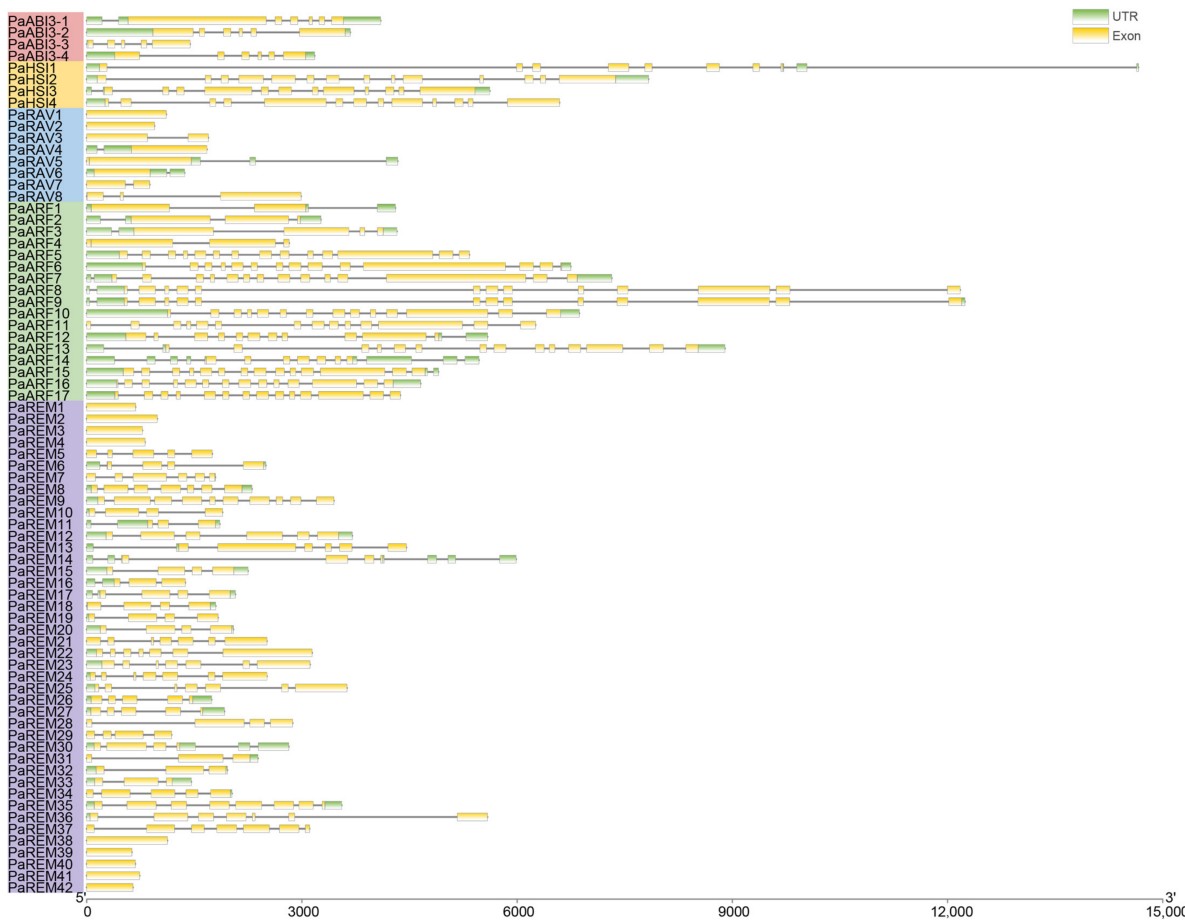

**Figure 2.** Exon–intron structures of apricot B3 genes. These structures were analyzed using TBtools. The exons and UTRs are represented by yellow boxes and green boxes, respectively. Five color blocks on gene names represent five subfamilies.

Using TBtools, 20 conserved motifs were identified from apricot B3 protein sequences (Figure 3). Some motifs were found in several subfamilies, such as motif 4, 11, and 14 in all subfamilies, except ARF. Members in the same subfamily had similar motif compositions. For instance, all of the HSI members had motifs 17, 14, and 4, and most of the RAV members had motifs 19, 11, 14, and 4. Motifs 1, 2, 9, and 5 were found in all ARF members, and their arrangements were similar. The motif composition of the REM subfamily was very complex, but it could still be observed that some members of the REM subfamily shared the same motif. Motifs 20, 10, 8, 18, 16, and 13 were specific to the REM subfamily.

Using NCBI Batch CD Search, important conserved domains on the apricot B3 protein sequences were identified (Figure 4). All members of the B3 superfamily had B3 domains, but the quantity and type of the domains in each subfamily were different. ABI3 members only contained one B3 domain. Three of the four HSI members had B3 and zf-CW domains. Five of the eight RAV members had B3 and AP2 domains, whereas the other three had only one B3 domain. PaRAV6 with a B3 and an AP2 domain was similar to the RAV subfamily rather than HSI, and this supported the phylogenetic tree by the ML method. All members in the ARF subfamily contained the B3 and Auxin_resp domain, and 11 members contained the B3, Auxin_resp, and aux_IAA domains. More than half of the REM members had multiple B3 domains.

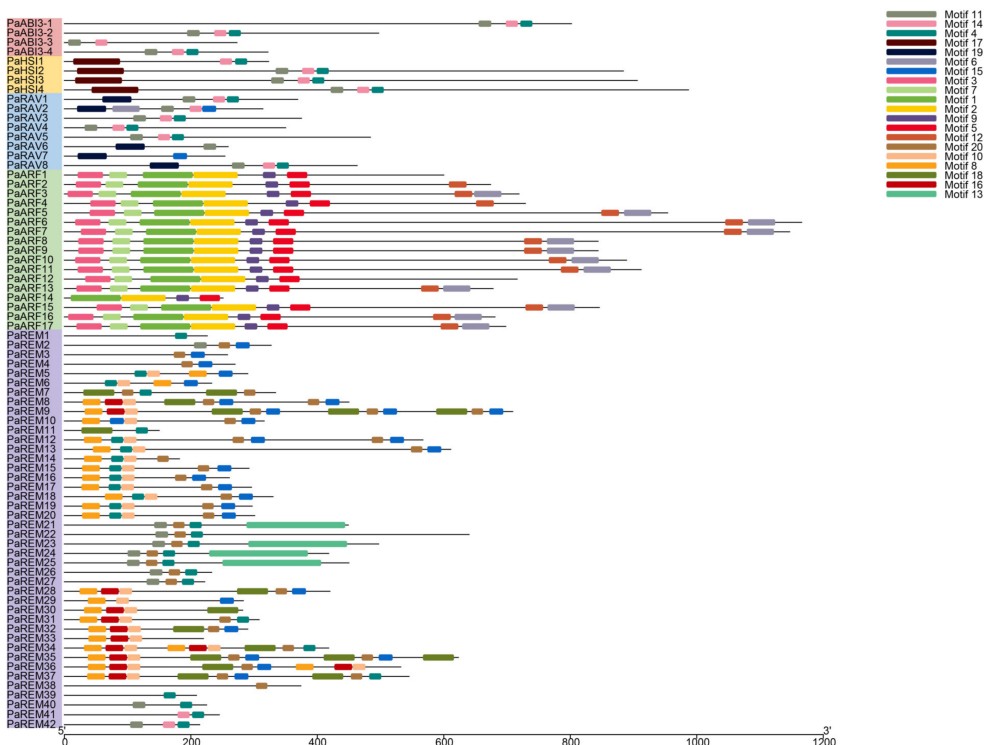

**Figure 3.** The conserved motifs of apricot B3 proteins. These conserved motifs were analyzed by using TBtools. A total of 20 types of color block on lines represent 20 types of motif. Five color blocks on gene names represent five subfamilies.

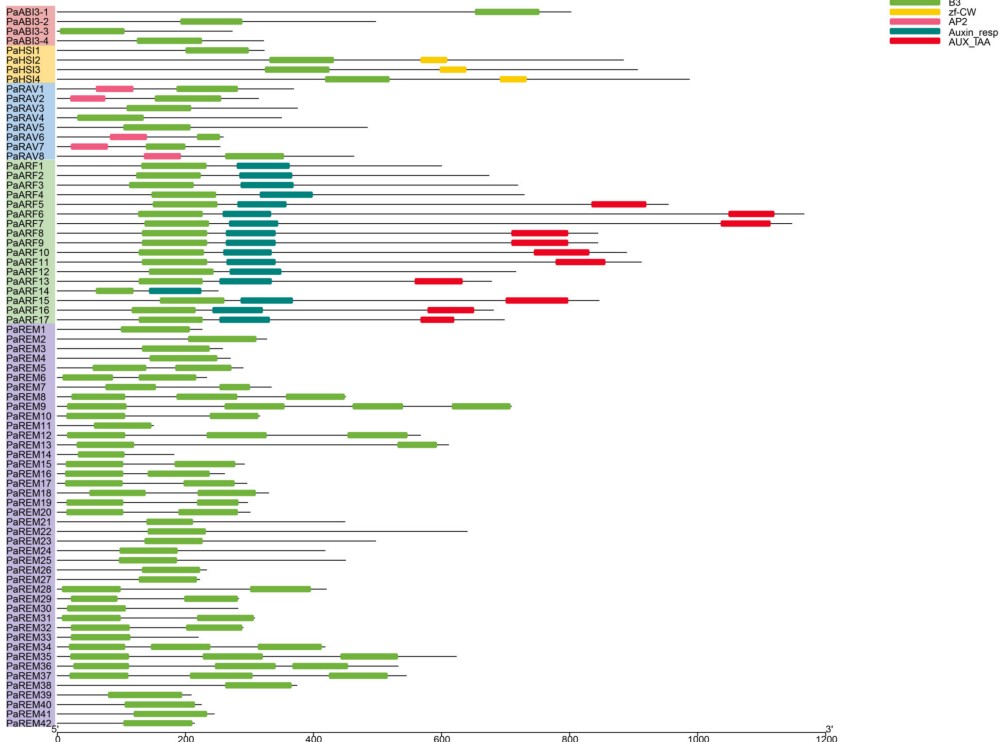

**Figure 4.** The conserved domains of apricot B3 proteins. These domains were identified by using NCBI Batch CD-Search and visualized using TBtools. A total of five types of color block on lines represent five types of domain. Five color blocks on gene names represent five subfamilies.

### 3.4. Chromosomal Locations and Synteny Analysis of Apricot B3 Genes

To determine the location of B3 genes on apricot chromosomes, Mapchart was used for visualization. All 75 genes are distributed on eight chromosomes (Figure 5). The number of genes on each chromosome was different, and the distribution of genes on the same chromosome was uneven. For instance, chromosome 7 had the most genes (18), and 15 of them were from REM subfamily, whereas the genes on chromosome 2 were the least (3) and were concentrated at the end of the chromosome.

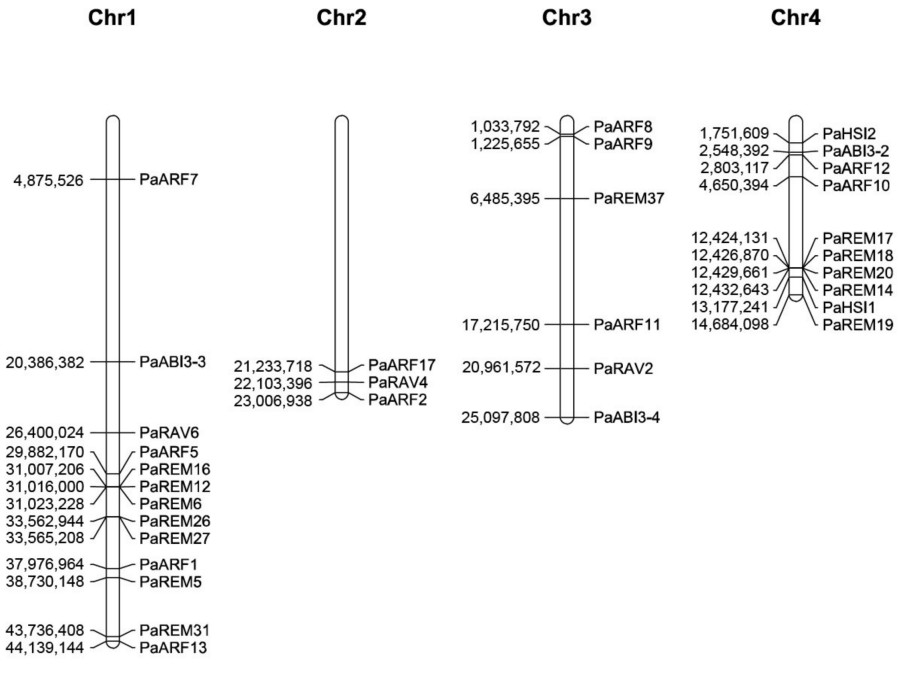

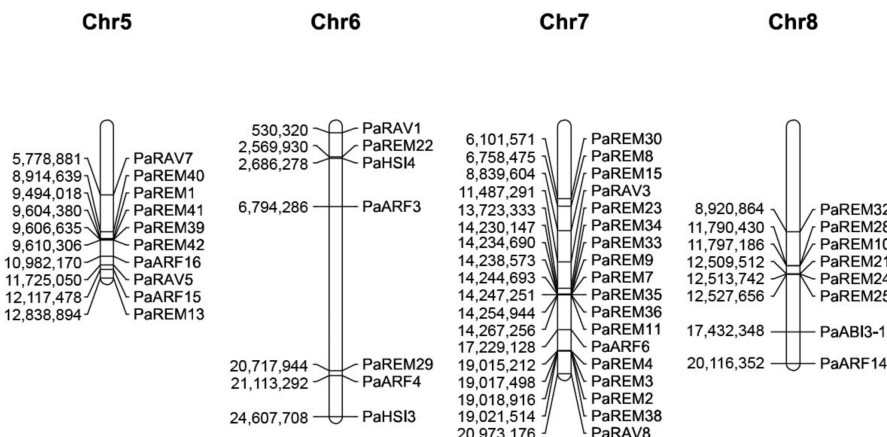

**Figure 5.** Chromosomal locations of the B3 genes on eight apricot chromosomes. The distribution of B3 genes on the chromosomes was visualized by using MapChart software. The right side of the chromosome displays the gene name, and the left side displays the position of this gene on the chromosome.

MCScanX was further used to check the duplication events of the B3 gene in apricot, and 12 segmental duplication events and 15 tandem duplication events were found (Figure 6a, Table S3). A total of 22 genes in the REM subfamily were involved in tandem duplication events.

In order to explore the evolutionary relationship of the B3 gene between apricot and other species, the syntenies of apricot and *Arabidopsis* were compared and analyzed, and 31 syntenic gene pairs were identified (Figure 6b, Table S4). In some cases, a single apricot gene was paired with multiple *Arabidopsis* genes. For example, *PaRAV6* formed a syntenic gene pair with *AT1G68840*, *AT1G13260*, *AT1G25560*, and *AT3G25730*, respectively.

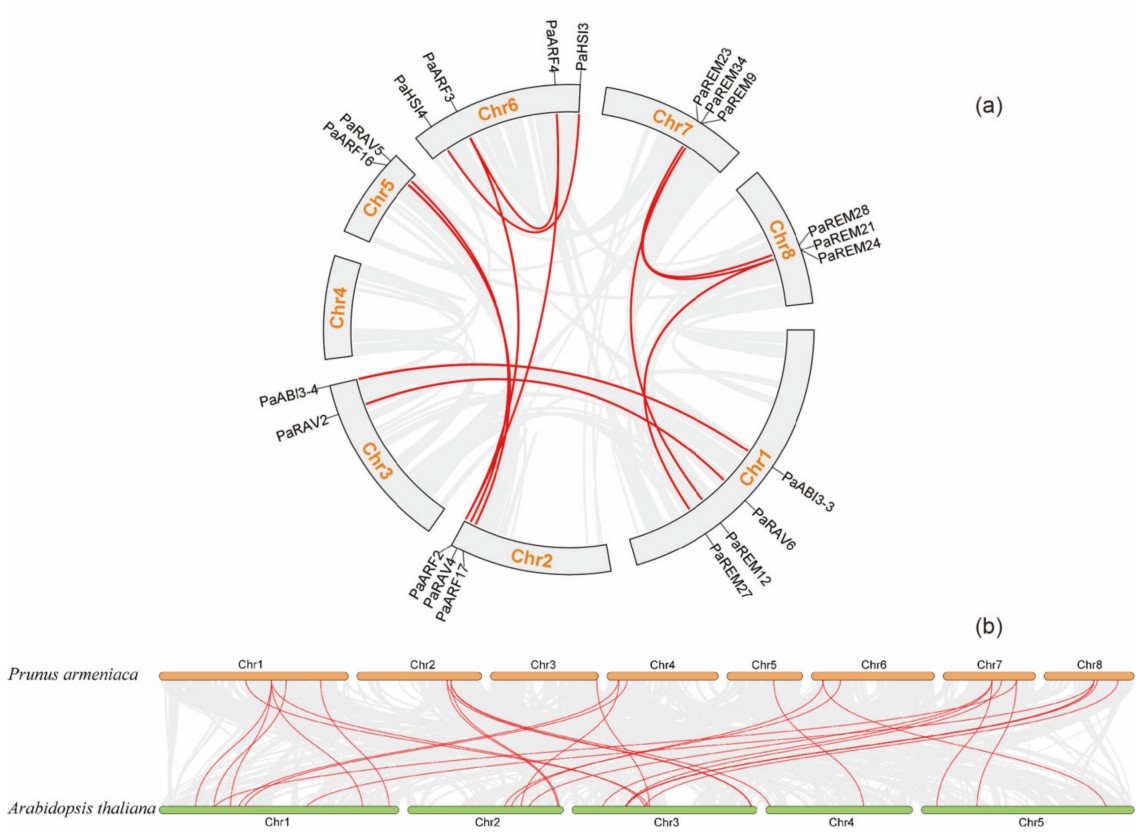

**Figure 6.** Synteny analysis of apricot B3 genes. (**a**) Synteny analysis of B3 genes in apricot. (**b**) Synteny analysis of B3 genes between apricot and *Arabidopsis*. The Multiple Collinearity Scan toolkit (MCScanX) was used to check the duplication events of B3 genes, and the Dual Synteny Plot in TBtools was used to plot these synteny figures. Gray lines show all segmental duplications, and red lines show syntenic B3 gene pairs.

### 3.5. Cis-Acting Elements and Gene Ontology (GO) Analysis of Apricot B3 Genes

To understand the potential role of apricot B3 genes, the 2 kb upstream sequence of each B3 gene was analyzed using PlantCARE (Figure 7). Numerous B3 genes have *cis*-elements related to plant hormones. For example, abscisic acid responsiveness elements are found in 55 B3 gene sequences, methyl jasmonate responsiveness elements are found in 50 sequences, and salicylic acid responsiveness elements are found in 50 sequences, etc. Many B3 genes are also associated with stress resistance. For instance, 65 B3 sequences contain stress-responsive elements, 33 contain low-temperature responsiveness elements, and 21 involve defense and stress responsiveness elements.

We also carried out GO enrichment analysis on apricot B3 genes, and the predicted functions mainly included the molecular function and the biological process (Figure 8). The prediction of molecular function showed that most B3 proteins were involved in DNA binding and transcriptional regulation. The prediction of biological process indicated that B3 protein played an important role in vernalization, flower development, and shoot system development.

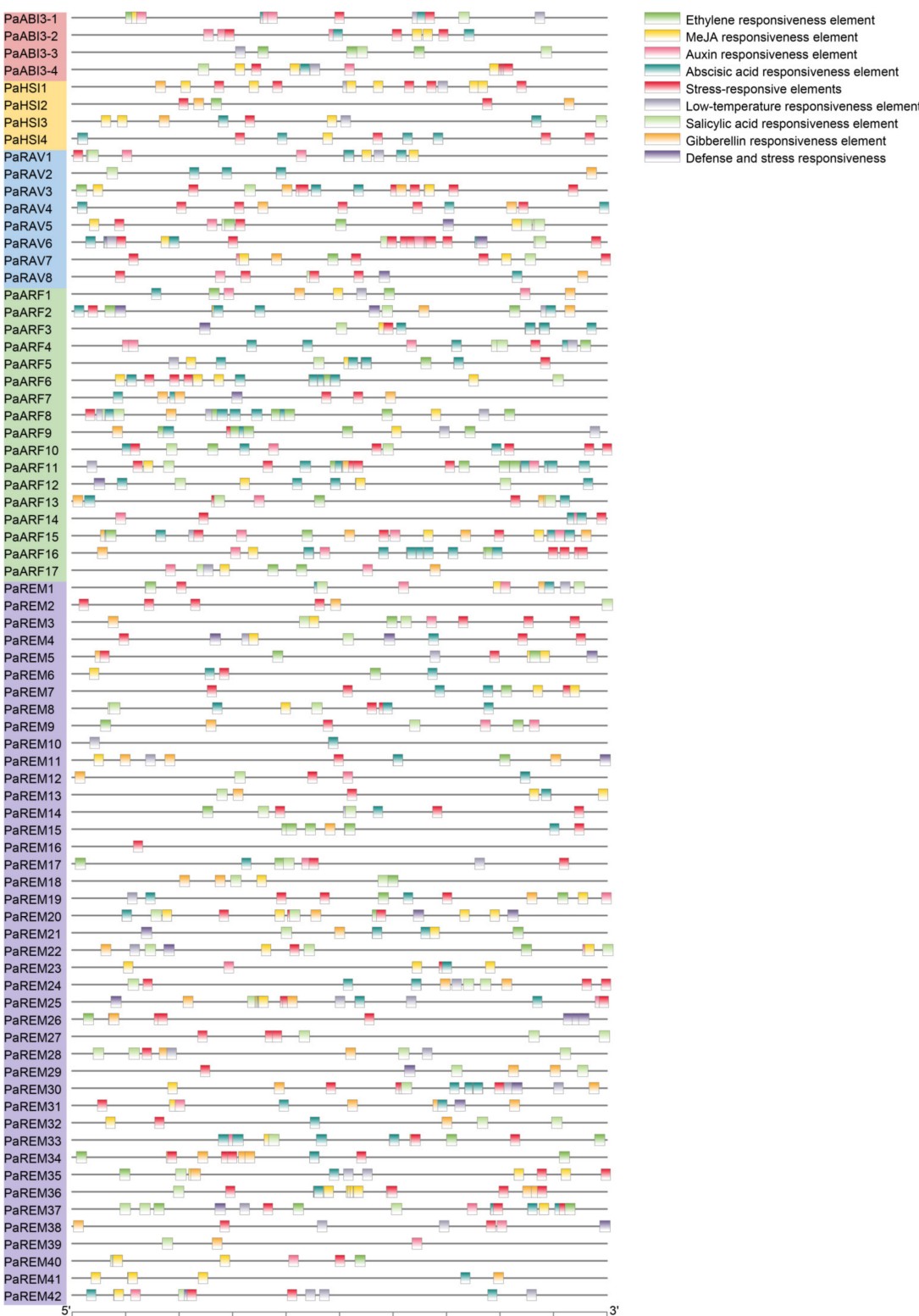

**Figure 7.** Predicted *cis*-elements related to hormones and resistance in apricot B3 gene promoters. These *cis*-acting elements were analyzed by using PlantCARE and were visualized using TBtools. The length of each promoter is 2.0 kb. A total of nine types of color block on lines represent nine types of *cis*-element. Five color blocks on gene names represent five subfamilies.

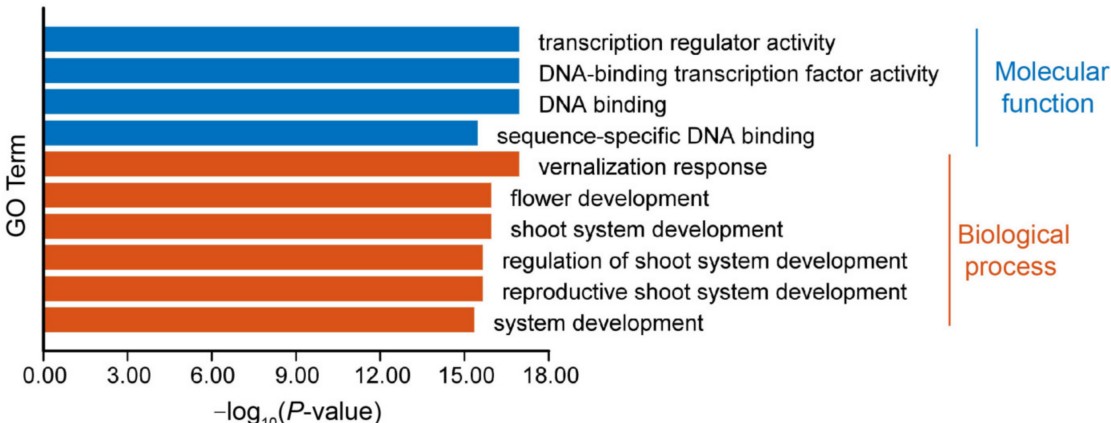

**Figure 8.** Gene ontology (GO) analysis of apricot B3 proteins. Gene functions were annotated using eggNOG-mapper, and GO enrichment was analyzed using TBtools. Blue and red bars indicate molecular function and biological process, respectively.

### 3.6. Tissue-Specific Expression Profiles and Identification of B3 Genes Associated with Oil Accumulation in Apricot

A total of 61 B3 genes showed expression profiles (FPKM > 1) in at least one tested sample (Figure 9). In total, 1, 5, and 10 genes were specifically expressed in flesh, flower or flower bud, and kernel, respectively, although most of these genes indicated an extremely low expression level. Most B3 genes displayed specific expression patterns across different tissues (Figure 9). Most of the ARF subfamily genes were more highly expressed in flesh and kernel than in flower and leaf, especially in the early development stages of the kernel. HSI subfamily members indicated higher expression levels in kernels than those in other samples, except *PaHSI2*. Interestingly, *PaHSI3* and *PaHSI4* exhibited an opposite expression pattern, of which the changing trends were gradually decreasing and increasing during kernel development, respectively. Most of the RAV subfamily members represented an extremely low expression level and had a relatively higher expression in flower bud or leaf. *PaRAV2* expressed in all samples except kernel, whereas PaRAV8 expressed specifically in kernel. *PaRAV3* expressed in the flower bud, leaf, and the late development of kernel. *PaRAV4*, *PaRAV5*, and *PaRAV6* represented the highest expression level in leaf. Most of the REM subfamily members indicated low expression levels in all tested samples, except for *PaREM12*, *PaREM15*, *PaREM18*, *PaREM28*, and *PaREM33*. Interestingly, *PaREM12* had a similar expression level in all tested samples and so did *PaREM28*. *PaREM33* was highly expressed in flower, and *PaREM15* was not expressed in leaf. *PaREM18* was specifically expressed in kernel and had a positive correlation with oil accumulation.

To identify ABI3 members that might have been involved in regulating oil accumulation in apricot and their relationships with *Oleosion*s, we conducted correlation analysis of the expression levels of *ABI3*s between oil contents and expressions of *Oleosin*s based on FPKM values, respectively (Table S5). Notably, most ABI3 subfamily members had a much higher expression level in kernel than in the other samples. *PaABI3-1*, *PaABI3-2*, and *PaABI3-4* indicated a positive correlation with oil accumulation. Through correlation analysis of expression levels between *ABI3*s and *Oleosin*s, we found that *PaOleosin1* (PaJ-TYG0200010520.01) with binding motif CATGCA had a high Pearson correlation coefficient (>0.9) with both *PaABI3-1* and *PaABI3-2*.

To verify the expression levels obtained from RNA-seq, we conducted RT-qPCR experiments using a sample during different kernel development (Figure S2). The correlation analysis of expression levels determined by RNA-seq and RT-qPCR indicated that our RNA-seq data have a high confidence.

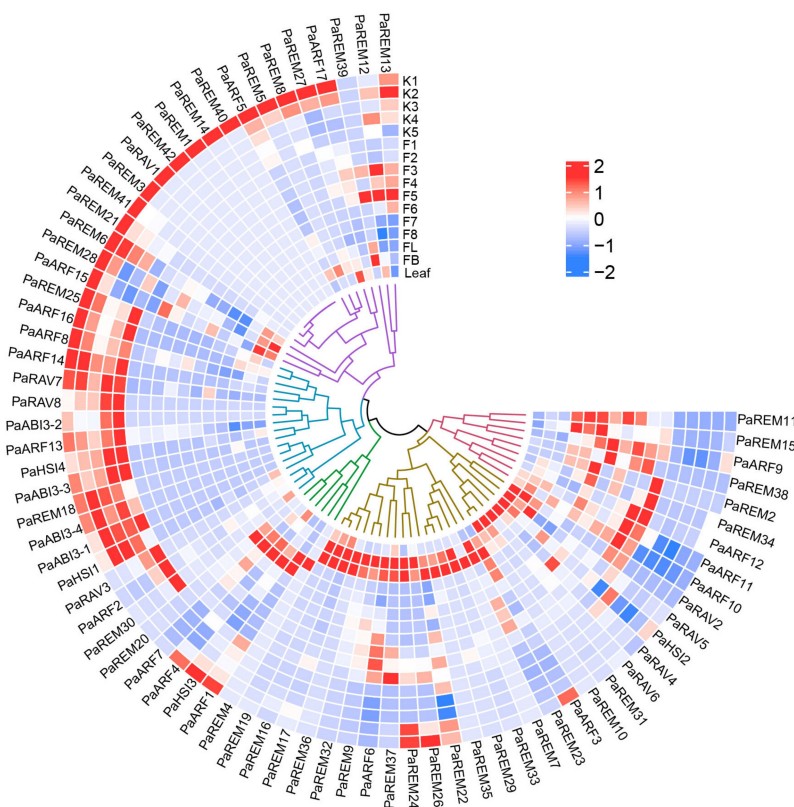

**Figure 9.** Gene expressions of apricot B3 genes in kernel, fruit, flower, flower bud, and leaf. Heat map was drawn based on Z-scores of FPKM. K1–K5: Kernels from different development stages. F1–F8: Flesh from different stage. FL: Flower. FB: Flower bud. Heatmaps were generated in the R package based on normalized FPKM values.

## 4. Discussion

The B3 transcription factor superfamily plays an important role in plant growth and development. In this study, we explore the structure and function of the B3 superfamily in *P. armeniaca*. In total, 75 B3 genes were identified from apricot by scanning its whole genome. The lengths of their corresponding proteins ranged from 150 to 1166 aa, similar to the length of B3 proteins in chickpea, tobacco, and pineapple [11,46,66].

Phylogenetic analysis is used to infer or estimate the relationship among individuals, providing a reliable method for exploring the relationship between sequence similarity and the functions of proteins belonging to the same family [67]. According to phylogenetic analysis, 75 apricot B3 genes were divided into five subfamilies (Figure 1). The members of the same subfamily were closely related and may have had uniform functions, which was similar to the situation observed in *Arabidopsis* and rice [20]. The REM subfamily contained the most genes, followed by ARF; the HSI and ABI3 subfamily had the least number of genes.

Comparative analysis of exon–intron organization is important for understanding the rules of gene structure and organization, protein functionality, and evolutionary changes among species [68]. Additionally, apricot B3 genes have 1–15 exons, which is similar to castor bean, tobacco, and grapevine [46,69,70], indicating the conservative nature of B3 genes in plants. We mapped the exon–intron structures of apricot B3 genes (Figure 2) and found that genes in the same subfamily had comparable structures, and the structures of genes clustered on the same branch within subfamily were more alike. For instance, ARF1–4 with 3–5 exons and ARF5–17 with 12–15 exons were clustered on two branches, respectively, which further verified the reliability of phylogenetic analysis. Exons encode amino acid sequences. So, proteins of the same subfamily have analogous functions. The difference in gene structure within subfamilies may be due to the occurrence of exon/intron gain/loss,

exonization/pseudoexonization, and insertion/deletion during gene duplication [71]. We also searched the conserved motifs and domains of apricot B3 transcription factors, and we found that their types and distributions were similar among members of the same subfamily (Figures 3 and 4). This result showed the conservation of motifs and domains in the subfamily, and it strongly supported phylogenetic analysis.

By chromosome localization of apricot B3 genes, we found that they were unevenly distributed on chromosomes and showed a phenomenon of aggregation (Figure 5). In particular, 15 genes of the REM subfamily were clustered on chromosome 7, and part of the remaining genes appeared in clusters in chromosomes 1, 4, 5, and 8. This situation, in which REM genes were closely linked on chromosomes, has also been observed in *Arabidopsis* and rice [20]. The existence of extensive gene duplication in the REM subfamily may have been the cause of clustering [46].

Gene duplication in the transcription factor family is likely to lead to the rearrangement of plant-specific protein domains, create new functions, expand the gene set, and ultimately drive plant morphogenesis to evolve. Some duplicated genes retain overlapping expression domains and functions, whereas others evolve new functions [72]. We checked the duplication events of apricot B3 genes, and we found 12 segmental duplication events and 15 tandem duplication events (Figure 6a, Table S3), indicating that tandem duplication is the main mode of apricot B3 family expansion. A total of 22 genes were in the tandem duplication, and they all belonged to the REM subfamily. In addition, one third of segmental duplication events happened in the REM subfamily. The large number of gene copies explained why the REM subfamily has plenty of members and complex structures. The research on the *Arabidopsis* B3 family also shows that the REM subfamily is less conserved than other subfamilies and that it has evident structural divergence [73]. This may lead to the diversification of REM gene functions.

The synteny analysis across species provides an in-depth understanding of the evolutionary relationship between different lineages, and it offers an opportunity to study the relationship between genome structure and the function of organisms [74]. We analyzed the synteny between apricot and *Arabidopsis*, and we identified 31 syntenic gene pairs (Figure 6b, Table S4). This indicates that many B3 genes originate from the common ancestor of *Arabidopsis* and apricot before lineage divergence. However, most apricot B3 genes have not been found to show homology in *Arabidopsis*. This may be caused by multiple chromosome rearrangements and fusions in the evolutionary process of apricot and *Arabidopsis* [75]. There is a phenomenon that one gene is paired with multiple genes in syntenic events, which occurs more frequently in apricot than in *Arabidopsis*. This result shows that an ancestor gene expanded fewer times in apricot but more times in *Arabidopsis* in the evolutionary process after speciation, which is accordance to the longer life circle of apricot. Normally, orthologs retain the same function in the course of evolution [76]. Therefore, we can infer the possible function of the B3 gene in apricot by observing the function of its ortholog in *Arabidopsis*.

*Cis*-acting elements present in the promoter region play an important role in regulating the gene expression of metabolic pathway-related genes [77]. Many identified *cis*-elements are involved in hormone responses and the transcriptional regulation of genes when encountering environmental stresses [78,79]. By analyzing the promoter sequences of apricot B3 genes, we identified *cis*-elements related to hormones and resistance (Figure 7), indicating that B3 genes were involved in the hormone regulation pathway and the resistance response to stress. Three of the four ABI3 subfamily members had auxin responsiveness elements, which indicated that ABI3 has functions in auxin action, and this was consistent with previous studies [80,81]. In addition, 16 of the 17 ARF subfamily members had abscisic acid responsiveness elements (ABRE), which indicated that ARF genes might participate in the ABA signaling pathway. The largest number of B3 genes are involved in stress response, which shows their potential in improving plant resistance. In particular, B3 genes containing low-temperature responsiveness elements deserve special attention. Apricot

is vulnerable to late frost, resulting in yield reduction or even no yield [82]. Therefore, B3 genes involved in low temperature response are of great significance to apricot.

Gene ontology (GO) is applied to the predictive tasks of functional genomics, focusing on the analysis of functional patterns related to gene products [83]. Through GO enrichment analysis of apricot B3 genes, we found that they have DNA binding and transcriptional regulation functions. In addition, they are closely related to vernalization, flower development, and shoot system development. These results are in line with the functions of B3 genes in previous studies. For instance, *VRN1* and *VAL1*, members of the *Arabidopsis* B3 superfamily, participate in vernalization [84,85]. The involvement of the REM subfamily in flower development has also been confirmed [18,27,86]. The ARF subfamily being regulated by auxin participates in shoot development [87]. The over-expression of *NGAL1*, a member of RAV subfamily in *Arabidopsis*, is capable of altering shoot development [88]. These studies show that the function of the B3 superfamily in plants is conserved and can provide a reference for further research of the B3 superfamily in apricot.

The expression levels of genes in different tissues and different developmental stages of plants reflect their functions. Some genes are specifically expressed in flesh, flower or flower bud, and seed, indicating that they play a role in specific tissues. The expression level of HSI subfamily members in kernel was higher than that in other tissues. Prior research has proclaimed that HSI genes can promote seed growth and development [89]. This may be the reason for their high expression levels in apricot kernel. The RAV subfamily members were more highly expressed in flower bud and leaf than those in flesh and kernel. Researchers have found that the RAV members *TEM1* (*RAV2-like*) and *TEM2* (*RAV2*) can prevent precocious flowering and postpone floral induction [90]. Overexpression of *RAV1* in *Arabidopsis* delays flowering, causes rosette leaf development slowing, and induces leaf senescence [91,92]. We speculate that the RAV subfamily may play an important role in the development of apricot flowers and leaves.

Apricot kernel oil has a rich fatty acid composition and contains bioactive compounds [93,94], which have great health benefits for people. Therefore, we pay special attention to B3 genes related to apricot kernel oil storage. *PaREM18* is specifically expressed in apricot seeds, and its expression level is positively correlated with oil accumulation. Its role in the accumulation of apricot kernel oil needs further research. Previous studies have shown that the ABI3 subfamily is involved in the seed development of oilseed crops, and that it promotes oil accumulation in seeds by upregulating LIPID DROPLET PROTEIN genes (including OLEOSINs and CALEOSINs) [30,95,96]. The regulation of ABI3 on the transcription of Oleosin, which encodes oil–body structural proteins, has been reported in many plant species, including *Arabidopsis thaliana* [96,97], *Brassica napus* [30], soybean [98], and castor bean [69]. As the lipid biosynthesis in plants are conservative, we mainly focus on the relationship between *ABI3* and *Oleosin* in apricot. In this study, we conducted correlation analysis on the expression level of *ABI3* and oil content, as well as the expression of *ABI3* and *Oleosin*, based on FPKM values. The results showed that *PaABI3-1*, *PaABI3-2*, and *PaABI3-4* were positively correlated with oil accumulation. The homolog of *PaABI3-1*, *PaABI3-2*, and *PaABI3-4* in *A. thaliana* was *AtABI3* (AT3G24650), *AtLEC2* (AT1G28300), and *AtFUS3* (AT3G26790), which belong to the ABI3/VP1 subgroup [19]. In *A. thaliana*, *ATFUS3*, *AtLEC2*, and *AtABI3* are often expressed specifically in developing seeds, participating in the regulation of seed oil accumulation [97,99]. These results suggest that certain B3 genes, especially *PaABI3-1*, *PaABI3-2*, and *PaABI3-4*, may play important roles in regulating oil accumulation in apricot kernel. In addition, *PaABI3-1* and *PaABI3-2* are highly positively correlated with *PaOleosin1* (PaJTYG0200010520.01). Thus, we speculated that *PaABI3-1* and *PaABI3-2* might play important roles in apricot by regulating the transcription of *PaOleosin*.

## 5. Conclusions

This study identified 75 B3 genes from apricot and divided them into five subfamilies (REM, ARF, ABI3, RAV, and HSI) through phylogenetic analysis. The exon–intron structures, conserved motifs, and domain compositions supported the results of phylogenetic

analysis. The B3 genes exhibited aggregation on apricot chromosomes, especially the REM subfamily. Tandem duplication was the main mode of apricot B3 family expansion, and gene duplication mainly occurred in the REM and ARF subfamilies. Many B3 genes originated from a common ancestor of *Arabidopsis* and apricot before lineage divergence, and ancestor genes expanded fewer times in apricot than in *Arabidopsis*. The apricot B3 genes are involved in numerous hormone regulatory pathways and resistance responses to stress. In addition to DNA binding and transcriptional regulation functions, the apricot B3 genes are also closely related to vernalization, flower development, and shoot system development. *PaABI3-1* and *PaABI3-2* might play a positive regulation role in the transcription of *PaOleosin*, which encodes a lipid body protein. Our research provides a reference for further explorations of the specific functions of the ABI3 subfamily in apricot kernel oil storage.

**Supplementary Materials:** The following supporting information can be downloaded at: https://www.mdpi.com/article/10.3390/f14081523/s1, Table S1: Primers used in this study. Table S2: Characteristics of apricot B3 transcription factor superfamily. Table S3: Duplication events of apricot B3 genes. Table S4: Synteny gene pairs of apricot\\*Arabidopsis*. Table S5: Correlation between the expression level of ABI3 and oil content, as well as ABI3 and Oleosin based on FPKM values. Figure S1: The phylogenetic tree of B3 family members constructed by NJ method. Five color blocks represent five subfamilies. The number on the tree branches represents the bootstrap value. Figure S2: RT-qPCR validation of the expression levels of B3 genes. The correlation between $\log_2$(fold changes) of B3 genes were analyzed by RNA-seq and by RT-qPCR. RT-qPCR results were expressed on the basis of *UBQ*.

**Author Contributions:** Conceptualization, H.L.; methodology, X.S., W.Y. and H.Z.; software, X.S., L.W. and W.Y.; validation, X.S. and L.W.; formal analysis, X.S.; investigation, X.S. and J.H.; resources, H.L.; data curation, X.S.; writing—original draft preparation, X.S.; writing—review and editing, X.S., W.Y., L.W. and H.L.; visualization, X.S. and T.W.; supervision, W.Y., L.W. and H.L.; project administration, W.Y. and H.L.; funding acquisition, H.L. All authors have read and agreed to the published version of the manuscript.

**Funding:** This research was funded by National Natural Science Foundation of China, grant number 32101562, and the central non-profit research institution of the Chinese Academy of Forestry, grant number CAFYBB2021MA008.

**Data Availability Statement:** The data presented in this study are available on request from the corresponding author.

**Conflicts of Interest:** The authors declare no conflict of interest.

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
