# Peer review of "Genome-Wide Identification and Expression Profiling of B3 Transcription Factor Genes in Prunus armeniaca"

_forests, doi:10.3390/f14081523_

Round 1

Reviewer 1 Report

The authors analyzed the characteristics of the B3 gene family, and revealed the important role of ABI3 subfamily in kernel development in P. armeniaca. The gene structure, phylogenetic relationship, and conserved motifs of B3 family genes, and the positions of them on the chromosomes were all clearly analyzed in this manuscript. And the expression patterns of these genes in different tissues and development stages were calculated by RNA-seq data. This paper could provide reference for the study of the B3 gene family in plants. There are some revise opinion as follows:

1.      Line 22, “apricot” would be replaced by “apricot (Prunus armeniaca)”

2.      Line 25, “B3 genes exhibit aggregation on 25 apricot chromosomes.”, this sentence is not clear.

3.      Line 155, cited the reference of (PRJCA001987), and supplied the methods of transcriptome data analysis to obtain the value of FPKM values in different tissues.

4.      Line 31-32 showed that PaABI3-2 and PaABI3-8 may play an important role in regulating apricot kernel oil storage, by reg-31 ulating the transcription of PaOleosin. But in the part of method, I could not find the methods to calculate the storage of kernel oil.

5.      In table 1, more information of the characteristics of apricot B3 transcription factor superfamily would be exhibited in this part. Such as Grand average of hydropathy, Aliphatic index, Instability index, etc.

6.      Figure 1, is not clear.

7.      Figure 2, the site of the domain would be supplied in the B3 genes.

8.      Figure 3, there is lack of the information of the motifs.

9.      Figure 4, the information of the marks at the left and right of the chromosomes would be remarked after the title of the figure.

10.   B3 genes are transcription factors, I could not clearly understanding the what is the significance of GO enrichment analysis?

11.   Line 288, Figure S1 would be cited in the main text, as part of Figure 8.

Reviewer 2 Report

I have reviewed your MS thoroughly.The data analyzed in the article is too simple, and the species used in the article are also very limited, and the basic contents of some gene families are not analyzed. The current analysis is not sufficient for publication in the current journal.My comments are as follows, hoping to help you with your research.

 Comment #1

The title of the article is not appropriate.The content of the article does not reflect that ABI3 subfamily plays important roles in kernel development and oil storage in Prunus armeniaca. This can only account for the differential expression in these tissues.

Comment #2

Grammar, style of English language and the sentence structure should be improved.

 Comment #3

 In Figure 2, yellow boxes represent exons, but CDS is shown in the figure. In addition, some of the figure legends in the text are too simple.

 Comment #4

There are some formatting problems with the text. In line 184 and 185, the Arabidopsis is non-italics. In line 244, the “cis-elements” is non-italics. And in line 237, the “plantCARE” was in lower-case letters.

 Comment #5

 What does the "55, 34, 40, 47, 49, and 48" in line 239 represent? It is not mentioned in the text.

 Comment #6

In line 424, there is no data indicating that PaABI3-2 and PaABI3-8 may play an important role in regulating apricot kernel oil storage by regulating the transcription of PaOleosin.

 Comment #7

The reference format is inconsistent, such as reference 25, 26, 27, 32, 34and so on. Please check on your reference format carefully and modify it.

The article needs to be polished

Reviewer 3 Report

Thanks a lot for such a good manuscript.

Would you please follow the following points to be able to publish your work.

Materials and methods:

This section is well-designed, but please mention the source where you obtain the “apricot”

Mention the model (name and number) of the RT-qPCR machine

Results:

In fig. 1, would you please mention the software applied to get this tree, in the figure caption.

In fig. 4, would you please mention the software applied to get these chromosomes, in the figure caption.

In fig. 6, would you please mention the software applied to get this promotor, in the figure caption.

Please apply this in all figures.

Patents

Please remove patent section from manuscript, as it is not offered

References:

Please follow the formatting of references, “bold” the year of the publication.

Reviewer 4 Report

Shi et al., identified B3 transcription factors in Prunus armeniaca, performed phylogenetic analysis of this gene family, compare gene structures, and investigated expression patterns of the B3 genes in different tissues. The manuscript is very well written, but details are missing in the methods and the conclusions are not fully supported by the results. Below are three major comments and a few minor suggestions: 

Major comments:

2.2 Multiple sequence alignments and phylogenetic analysis

  • Instead of using NJ, which is a clustering algorithm, maximum likelihood or Bayesian methods should be used to infer phylogeny. Both methods are implemented in MEGA and should be easy to execute. The results from those three methods may not differ much, but to tell a phylogenetic story, the correct method must be used. 
  • I would also suggest to add additional genomes in the phylogenetic analysis, which may help better resolve the tree and provide insights of the evolutionary history of this gene family, especially the ABI3 clade.

3.2 Phylogenetic analysis of apricot B3 genes

According to the phylogeny, ABI3 genes are not in one monophyletic group. What is the justification of assigning them into the same subfamily? You can flip the HSI clade and the ABI3 clade next to it (which contains PaABI3-8 to PaABI3-10) and the tree remain the same, but the red color block will be broken into 2 groups.

Line 155: How exactly is the RNAseq mapping and quantification performed? Please include a brief description and the reference paper. Additionally, FPKM should not be used to compare expression across samples, please use TPM or TMM.

Minor comments:

Line 111: ‘the correlation between ABI3 and oil content’ is unclear. Is it the expression level of ABI3? Please clarify. 

Line 119: I would suggest to add the cultivar information after P. armeniaca as well as more and more cultivar specific genomes are being sequences, it might be confusing which specific genome was used in this study in the future.

Line 120: Which version of Arabidopsis annotation was used? TAIR10? Araport11? Please specify. 

Line 155: I didn’t find RNA-seq information under PRJCA001987. Please provide a direct link to the data. Is there replications for each sample? If so, how many? 

Lines 171-172 and section 3.3: is the variation in protein size and exon numbers expected? Since the authors mentioned that this gene family has been studied in peach, how does those compare to B3 genes in peach? Is it possible that the annotation of some of the B3 genes from the apricot genome are erroneous?

Figure 2: Figure legends are too small and almost invisible while printed out. I’d suggest to use boxes or shades to distinguish the 5 subfamilies. What does upstream/downstream mean? Are those UTR regions identified in the genome annotation?

Figure 3 and 4: Same as figure 2, please add boxes or shades or other types of indicator to separate the 5 subfamilies. 

Section 3.5: why is only the 2kb upstream was selected in the analysis? Please justify or add reference for support.

Reviewer 5 Report

- Cite more newly-published references.

- I suggest the introduction should be start with the Aprocot introduce, continued with problem statement and importance of the B3 transcription factor.

- Please add more information about the assessed genes in Table S1 including product length and gene IDs.

- In the section 2.6, the RNAseq analysis should be extent with more details.

- Add procedure of correlation calculation (Fig. S1) in the material and methods.

- Mention to the Genome Reference version which you used in the analyses, for example in section 2.4.

- I could not see the related graph/table of the following result:

"We conducted correlation analysis on ABI3 and oil content, as well as ABI3 and Oleosins, based on FPKM values."

Round 2

Reviewer 2 Report

The authors have addressed all my comments and accordingly revised the manuscript. But the article also has minor methodological errors and text editing to revise, my comments related to improving the article are as follows:

1. The figures need to be adjusted to high resolution pictures, and it looks very blurry.

2. Some of the reference formats have underlining, which needs to be removed.

3. Table 1 is recommended for inclusion in the supplementary materials.

4. In line 151, the reference 56 position is suggested to be placed at the end of the sentence (PlantCARE (http://bioinfor- 152matics.psb.ugent.be/webtools/plantcare/html/) [56]).

The grammar of the article is suggested to be revised.
